# Impact of Multidisciplinary-Led Implementation of Antimicrobial Stewardship Programs in Zambia: Findings and Implications

**DOI:** 10.3390/antibiotics14111125

**Published:** 2025-11-07

**Authors:** Joseph Yamweka Chizimu, Steward Mudenda, Victor Daka, Webrod Mufwambi, Zoran Muhimba, Kaunda Yamba, Misheck Shawa, Kelvin Mwangilwa, Jimmy Hangoma, Sombo Fwoloshi, Amon Siame, Kaunda Kaunda, Andrew Bambala, Kenneth Kapolowe, Priscilla Nkonde Gardner, Duncan Chanda, Shempela Doreen, Charles Chileshe, Paul Simujayang`ombe, Ntombi B. Mudenda, Loveness Sakalimbwe, Aubrey C. Kalungia, Chikwanda Chileshe, Taona Sinyawa, Maisa Kasanga, Raphael Chanda, Samson Mukale, Shikanga O-Tipo, Evelyn Wesangula, Cephas Sialubanje, Adrian Muwonge, Fred Simwinji, Chie Nakajima, Freddie Masaninga, Fusya Goma, Nyambe Sinyange, Yasuhiko Suzuki, John Bwalya Muma, Roma Chilengi

**Affiliations:** 1Antimicrobial Resistance Coordinating Committee (AMRCC), Zambia National Public Health Institute, Lusaka 10101, Zambia; steward.mudenda@unza.zm (S.M.); zmuhimba@yahoo.com (Z.M.); mwangilwakelvin@yahoo.com (K.M.); priscillagardner82@gmail.com (P.N.G.); doreenshempela@gmail.com (S.D.); chichalesi2@gmail.com (C.C.); paul.simujayangombe@znphi.gov.zm (P.S.); lovesssakalimbwe@gmail.com (L.S.); chikchile@gmail.com (C.C.); kasangamaisa@yahoo.co.uk (M.K.); mukalesamson14@gmail.com (S.M.); sialubanje@gmail.com (C.S.); bsinyange@gmail.com (N.S.); chilengr@yahoo.com (R.C.); 2Department of Pharmacy, University of Zambia, Lusaka 10101, Zambia; webrod.mufwambi@unza.ac.zm (W.M.); ckalungia@unza.ac.zm (A.C.K.); 3Department of Public Health, Copperbelt University, Ndola 10101, Zambia; dakavictorm@gmail.com; 4University Teaching Hospitals, Lusaka 10101, Zambia; sombofwoloshi@gmail.com (S.F.); bambalaandrew@gmail.com (A.B.); duncanchanda@gmail.com (D.C.); 5Action on Antibiotic Resistance (ReAct) Africa, Lusaka 10101, Zambia; kaundayamba@gmail.com (K.Y.); raphael.chanda@reactafrica.org (R.C.); 6Hokudai Center for Zoonosis Control in Zambia, Hokkaido University, Lusaka 10101, Zambia; misheckshawa@czc.hokudai.ac.jp; 7Department of Pharmacy, Levy Mwanawasa Medical University, Lusaka 10101, Zambia; jimmyhangoma0282@gmail.com; 8Centres for Infectious Disease Research in Zambia, Lusaka 10101, Zambia; amon.siame@cidrz.org (A.S.); kaunda.kaunda@cidrz.org (K.K.); fred.simwinji@cidrz.org (F.S.); 9Levy Mwanawasa University Teaching Hospital, Lusaka 10101, Zambia; kennethkapolowe2020@gmail.com; 10School of Veterinary Medicine, University of Zambia, Lusaka 10101, Zambia; ntombi.nkonde@unza.ac.zm (N.B.M.); jmuma@unza.ac.zm (J.B.M.); 11Department of Veterinary Services, Ministry of Fisheries and Livestock, Lusaka 10101, Zambia; taonasinyawa@gmail.com (T.S.); fusyagoma@gmail.com (F.G.); 12Department of Health, World Health Organization, Lusaka 10101, Zambia; otipos@who.int (S.O.-T.); masaningaf@who.int (F.M.); 13Strengthening Pandemic Preparedness, Eastern and Southern Africa Health Community, Arusha 1009, Tanzania; ewesangula@ecsahc.org; 14Old College, University of Edinburgh, South Bridge, Edinburgh EH8 9YL, UK; adrian.muwonge@roslin.ed.ac.uk; 15Division of Bioresources, Hokkaido University International Institute for Zoonosis Control, Sapporo 001-0021, Japansuzuki@czc.hokudai.ac.jp (Y.S.); 16Division of Research Support, Hokkaido University Institute for Vaccine Research and Development, Sapporo 001-0021, Japan

**Keywords:** antimicrobial stewardship, antimicrobial resistance, healthcare facilities, WHO Core Elements, Zambia

## Abstract

**Background/Objectives**: Antimicrobial resistance (AMR) is a critical global health threat, with sub-Saharan Africa disproportionately affected. Antimicrobial stewardship (AMS) programs are essential in combating AMR; however, data on their implementation in resource-limited settings like Zambia remain scarce. This study assessed the post-implementation status of World Health Organization (WHO) AMS core elements in selected public hospitals in Zambia. **Methods**: A cross-sectional pre- and post-intervention survey was conducted in 11 public hospitals across Zambia’s 10 provinces. Baseline (pre-implementation) and 12-month follow-up (post-implementation) assessments were carried out using the WHO-adapted Periodic National and Healthcare Facility Assessment Tool. The six AMS core elements evaluated included leadership, accountability, AMS actions, education, monitoring, and feedback. **Results**: The average AMS program score increased from 59% at pre-implementation to 81% at post-implementation. Significant improvements were observed in education and training (+36%) and accountability (+31%). While leadership and monitoring also showed positive trends, gaps persisted in AMS actions (63%) and feedback/reporting mechanisms (68%). Drug and Therapeutics Committee (DTC) functionality improved by 23%, with 90% of facilities now holding regular DTC meetings. Implementation of AMS actions, such as ward rounds, rose from 0% to 73%. Challenges remained in clinical audit feedback, resource mobilization, and prescribing optimization. Variability across facilities highlighted differences in leadership, resources, and technical capacity. **Conclusions**: AMS implementation in Zambia improved substantially across key domains. However, sustained leadership, adequate financing, and continuous capacity-building are needed to address persistent gaps and ensure long-term success in mitigating AMR.

## 1. Introduction

Antimicrobial resistance (AMR) is a critical and escalating global public health challenge [1,2,3]. It was estimated that more than one million deaths annually were directly attributable to AMR, with an even greater number of deaths indirectly associated with resistant infections between 1990 and 2021 [4,5]. If AMR is not addressed, then more than 39 million deaths will occur due to AMR [5,6]. Sub-Saharan Africa bears a disproportionate share of this burden, with an estimated 23.5 deaths per 100,000 population linked to AMR, among the highest rates globally [7,8].

The *Monitoring Antimicrobial Resistance and Antimicrobial Use* (MAAP) project, which assessed AMR trends across 14 African countries, revealed alarmingly high resistance levels in all participating nations [9]. Notably, Zambia recorded a Drug Resistance Index (DRI) of 60.8%, one of the highest in the region, with fluoroquinolones and third-generation cephalosporins demonstrating resistance levels exceeding 60.9% during the 2016–2018 period [9]. National AMR surveillance data from Zambia similarly showed resistance rates exceeding 80% to third-generation cephalosporins among common pathogens such as *Klebsiella pneumoniae* and *Escherichia coli*, particularly in tertiary care settings [10]. These findings highlight the urgent need for context-specific interventions to curb AMR and preserve antibiotic efficacy.

One of the cornerstone strategies to combat AMR is the implementation of antimicrobial stewardship (AMS) programs [11,12,13,14]. AMS is defined as a coordinated set of interventions designed to promote the appropriate use of antimicrobials, improve patient outcomes, reduce AMR, and decrease unnecessary healthcare costs [15]. Key components of AMS include prescriber education, infection prevention and control (IPC), audit and feedback mechanisms, antimicrobial use surveillance, and policy development [16,17]. Effective implementation of AMS has been reported to have a positive impact on the appropriate use of antibiotics, improved awareness and knowledge of antimicrobial use (AMU) and AMR [18,19,20,21,22,23,24].

Implementing AMS programs in low- and middle-income countries (LMICs) is hindered by numerous systemic, resource, and capacity-related challenges. For instance, a study conducted in Malawi found that implementation of AMS programs in hospitals faced weak leadership commitment, inadequate resource allocation for AMS activities, limited or absent AMS ward rounds and prescription audits in most facilities, and fragmented education and training [25]. Another study conducted in 15 public hospitals in two regions of Ghana using the WHO AMS core elements toolkit revealed suboptimal performance across most domains, particularly leadership commitment, pharmacy expertise, implementation of AMS interventions, and monitoring of antibiotic use and resistance rates [26].

AMR has been reported as a public health problem in Zambia, with most microbes found to be resistant to conventional antimicrobials [10]. Evidence has found high resistance of microbes against critical antimicrobials used in human health [27,28,29,30,31]. The AMR problem has been worsened by gaps and challenges in laboratory capacity to conduct microbiology and diagnostic stewardship [9,32,33]. Further, there are gaps in the implementation of AMS programs across Zambian hospitals as reported in a previous study [34]. These gaps and challenges affect the effective implementation of AMS programs and patient outcomes.

This study presents findings from the post-implementation assessment of AMS programs in 11 public hospitals across Zambia’s 10 provinces, using a WHO-modified Periodic National and Healthcare Facility Assessment Tool. The study aimed to provide evidence on the effectiveness of multidisciplinary-led AMS implementation and inform future strategies to strengthen AMS and combat AMR in Zambia and similar low-resource settings. The primary aim of the study was to assess changes in AMS core element implementation after interventions, and the secondary aims were to identify persistent gaps and facility-level variations in Zambia.

## 2. Results

This study assessed a mix of tertiary and secondary public hospitals across six provinces of Zambia, with bed capacities ranging from 230 to 741 and annual admissions from 3500 to nearly 20,000 (Table 1). Tertiary hospitals, particularly in the Copperbelt Province, had the largest bed capacities and patient loads, reflecting their role as major referral centers. Secondary hospitals, such as Mansa General and Chilonga Mission General, had smaller capacities and lower admission volumes, highlighting potential differences in infrastructure, resource availability, and service delivery capacity between levels of care (Table 1).

Most participants were female (63%), while males constituted 38% of the sample. Pharmacists and clinicians were the most represented professional groups, each accounting for 25% of respondents. Laboratory personnel made up 22%, nurses 19%, and environmental technologists 9% (Table 2). This distribution shows a diverse mix of healthcare professionals involved in the study, with some variation in representation across professions.

### 2.1. Changes in the AMS Program Implementation Between the Baseline and Period One Assessment

There was an overall improvement in the assessment scores for the period one assessment (average score of 81%) when compared to the baseline assessment (average score of 59%) across all the core elements. More improvements were observed in the core elements Education and Training (% difference of 40%) and accountability and responsibility (% difference of 26%). While marginal improvements were also observed in the core elements reporting feedback and Monitoring and Surveillance, with percentage differences of 22% and 14%, respectively. Though improvements were observed, hospitals had average scores below 80% in 3/8 of the core elements in period one assessment, which included DTC functionality status (average score = 72%), AMS actions (average score = 70%), and reporting and feedback (average score = 70%), Figure 1.

Poor scores (below 50%) in hospitals were recorded for core elements; AMS actions for CnGH (45%) and KGH (45%) in reporting feedback in the period one assessment (Figure 2).

Overall, among the hospitals assessed, CTH had the highest (97%), followed by NTH (92%), while KCH and CGH had the lowest, 70% and 74%, respectively. Nevertheless, CGH and LGH had marked increases in their scores for the period one assessment compared to their baseline results of +35% and +43%, respectively (Figure 2).

Additionally, the core elements in which healthcare facilities performed poorly in the baseline assessment, including DTC functionality status, education and training, AMS actions and reporting, and feedback, were further analyzed on the period one assessment scores (Figure 3, Figure 4 and Figure 5).

### 2.2. DTC Functionality Status

There was a positive difference between the baseline and period one assessment average scores of 23%. All the hospitals indicated improvements in the assessed indicators. CCH scored the highest (90%), followed by KGH, which scored 80%. KTH and SGH hospitals scored the lowest on DTC functionality, with scores of 58% and 60%, respectively. Ninety percent (9/11) of the hospitals fully implemented the regular holding of the monthly DTC meetings and reported their performance activities to management. However, only 21% (2/11) had a medicine use policy and procedures, with 50% (5/11) having developed action plans approved by management. Thirty percent (3/11) conducted supply and medicine use problem studies and reported their performance activities to management (Figure 3).

### 2.3. Leadership Commitment

There was an improvement in both the average score and full implementation, from 57% to 80% and from 28% to 56%, respectively (Appendix A). All hospitals showed an increase in the average scores in period one assessment, except for MGH and CnGH. Fully implementation of above 80% among the hospitals was observed in indicators; AMS was identified as a priority by management (9/11, 82%), and having pharmacists dedicated to stewardship (11/11, 100%). However, most of the hospitals did not fully implement the following indicators: having facility annual action plans with key performance indicators (3/11, 18%), management allocating human and financial resources to AMS activities (physicians dedicated to stewardship (7/11, 64%)), facility action plan in place that prioritizes AMS activities (7/11, 64%), mechanisms to regularly monitor and measure implementation of AMS activities (6/11, 55%). Only 2/11, 18%, fully implemented the indicator of having a dedicated financial support for the AMS action plan. Additionally, on 4/11, 36% of hospitals had the budget for the implementation of the AMS action plan developed. (Appendix A).

### 2.4. Accountability and Responsibility

Generally, all the hospitals except KGH showed an improvement across the indicators in period one (average 87%) when compared to the baseline assessment (average 61%). Of the 11 indicators, full implementation among the hospitals was above 80% in the following four indicators: having a dedicated AMS champion, other healthcare professionals apart from the AMS team being involved in AMS activities, having an AMS coordinating unit on IPC programs and collaborating with other healthcare teams such as drug and therapeutics. Nevertheless, not all hospitals at the period one assessment fully implemented the indicators on AMS committee leadership having clear terms of reference (7/11, 64%), committee meeting on a regular basis (7/11, 64%), the champion for AMS having dedicated staff time for AMS activities in their job description (6/11, 55%), produce regular activity report on AMS implementation (7/11, 64%) and dissemination of AMS activity report to management and other team members (5/11, 45%).

### 2.5. AMS Actions

The average score across all AMS action indicators increased from 52% at baseline to 70% at the Period One assessment. Correspondingly, the percentage of hospitals fully implementing the assessed AMS actions rose from 30% to 51%. At baseline, only a minority of hospitals, 18%, 2/11, had established standard treatment guidelines with full implementation when compared to 36%, 4/11, during Period One. However, the periodic review and update of these guidelines remained poorly implemented, with only 9%, 1/11 of hospitals achieving full compliance in both assessments. The practice of conducting regular reviews or audits of antibiotic therapy improved from 18%, 2/11 at baseline, to 36%, 4/11 in Period One. Similarly, the accessibility of feedback from AMS teams to prescribers improved, with fully implemented feedback mechanisms increasing from 36%, 4/11, to 55%, 6/11. Significant progress was observed in the implementation of ward rounds and AMS team-led interventions. At baseline, none of the hospitals (0%) conducted regular AMS ward rounds or interventions, whereas by Period One, 73%, 8/11 of hospitals had incorporated these practices.

Further, access to laboratory and imaging services capable of supporting AMS interventions increased from 73% to 91%. Information technology (IT) systems, including decision-support tools, reached full implementation (100%) during Period One. Additionally, the proportion of hospitals utilizing standardized prescription charts, medical records, and transfer notes to support AMS activities increased from 45% at baseline to 64% in Period One. However, no improvement was noticed in hospitals having written policies requiring prescribers to document the clinical indication and the antibiotic prescribed. Across the hospitals, KTH, NTH, CCH, and SGH exhibited high scores of 80% and above, whereas CnGH, MGH, and KGH recorded a decrease or no progress in the average scores Figure 3.

### 2.6. Monitoring and Surveillance of AMR in Surveyed Hospitals

Most hospitals performed well on the indicators, such as healthcare facilities’ ability to regularly monitor shortages of essential antimicrobials (91%, 10/11 hospitals fully implemented) and the presence of mechanisms to report substandard and falsified medicines and diagnostics (100% of hospitals fully implemented). Additionally, there was an increased percentage of full implementation among the hospitals (from 52% to 70%); only 2/11 hospitals (LGH and SGH) had average scores below 80% when compared to the others in the period one assessment (Appendix A).

### 2.7. Education and Training

Under the education and training core element, healthcare facilities 7/11 scored above 80% in period one assessment when compared to baseline (2/11). While the other hospitals improved in terms of overall performance, KGH indicated a drop. Of the three indicators under this core element, 82%, 9/11 of hospitals fully implemented the training of the AMS team on AMS/IPC at the Period One assessment. Despite the improvements, only 27%, 3/11 and 64%, 7/11 of hospitals fully implemented AMS programs, such as optimizing antibiotic prescribing and offering continuous in-service training on AMS, respectively (Appendix A).

### 2.8. Reporting and Feedback Within the Healthcare Facility

All the healthcare facilities showed improved scores on the indicators for reporting and feedback (Figure 6). 5/11 at period one compared to 2/11 at baseline of the healthcare facilities, fully implemented the indicator on the AMS teams, communicating findings from audit/reviews of the quality/appropriateness of the antibiotic use to prescribers, along with specific action points. Marginal improvements were also observed in healthcare facilities developing and aggregating antibiograms and regularly updating them (4/11 at baseline versus 5/11 at period one) (Figure 4). On the indicator that described systems linking monitoring and reporting of healthcare-associated infections (HAIs), antimicrobial use, AMR, patient outcomes, and quality of care, 3/11 healthcare facilities fully implemented this indicator both at baseline and period one assessments. Nevertheless, more healthcare facilities (3/11) partially implemented this indicator at period one when compared to 0/11 at baseline assessments.

## 3. Discussion

This study demonstrated significant progress in the implementation of (AMS) Programs across the eleven (11) surveyed Zambian public healthcare facilities. The average assessment scores increased from 59% at baseline to 81% in the first follow-up period, indicating growing institutional commitment and heightened awareness of AMR as a national public health priority [35,36]. Though improvements were evident across all the WHO AMS core elements, disparities in performance among facilities and specific elements persisted, necessitating targeted support and sustainable interventions. Facilities such as CCH, NTH, and LTH maintained or improved their high scores, reflecting the effective implementation of stewardship, sustained institutional support, and strong engagement from the AMS team and, in some cases, external technical or financial assistance. Similar results have been reported in other countries, including Kenya, Tanzania, Uganda, and Ghana, where significant improvements were seen across the core elements such as governance and accountability [16,37]. In contrast, hospitals like KCH and MGH experienced declines or continued to underperform, suggesting systemic challenges that undermine AMS momentum. These may include staff attrition due to frequent transfers and weak administrative support. Variability between facilities pointed to local operational differences, resource disparities, and uneven engagement in stewardship programs. Identifying these gaps offers opportunities for targeted capacity-building, peer-to-peer mentorship, and the dissemination of best practices to foster more equitable and sustained AMS implementation across the healthcare facilities.

Significant improvements were observed across several key AMS components. Education and training recorded the largest gain, increasing by 41 percentage points, followed by accountability and responsibility, which increased by 26 percentage points. Leadership commitment also improved markedly, rising from an average of 57% to 81%. By the first follow-up, 82% of hospitals had prioritized AMS, and 100% had pharmacists dedicated to stewardship. These findings are consistent with studies from other African contexts, where leadership engagement and AMS champions have been pivotal to program success [38,39,40]. The improvements mirrored gains seen in Uganda and South Africa following pharmacist-led AMS training initiatives [41,42]. Despite progress, only a minority of hospitals had fully integrated AMS into annual institutional action plans or secured dedicated financial support. This was because of other demanding and competing hospital priorities for the available government resources [43]. These gaps echoed findings from LMICs, where AMS initiatives often relied on donor funding and lacked domestic financial planning [44,45], undermining sustainability, indicating institutionalization challenges.

In this study, while some hospitals reported incorporating AMS into planning processes, full implementation lagged. Inadequate implementation of AMS programs has been reported in other African countries [26,34,38,46]. Our study found that some barriers to the full implementation of AMS programs in healthcare facilities included insufficient guidance, conflicting priorities, administrative delays, or inadequate leadership commitment. Additionally, persistent deficiencies in developing medicine use policies and action plans suggested broader challenges in integrating evidence-based policy into clinical practice, a recurring issue in many African countries, which affects patient outcomes [38,44,47]. Although accountability and responsibility structures improved markedly (61% to 87%), clinical stewardship practices lagged. AMS actions and reporting, and feedback remained underperforming, with average scores of 70% each at the period one assessment. Our study also found that DTC performance improved significantly, with a 23% average increase. Alongside this, most hospitals conducted regular DTC/AMS meetings and reported performance activities to management. The improvements in the implementation of AMS components reported in our study could be due to the strengthened AMS activities and education across the country. These findings have been reported in other studies on the impacts and benefits of implementing AMS programs in healthcare facilities [18,48]. AMS interventions have been found to be influential in promoting the rational use of antimicrobials, awareness, knowledge, AMU, and combating AMR [38,49,50,51,52]. In areas of inadequate leadership commitment in implementing AMS programs, targeted training can be instigated for the hospital leadership [19,53,54,55,56].

Monitoring and surveillance capacity improved, with average scores increasing from 74% to 88%. These findings are similar to the findings from previous studies [38,57,58,59]. Notably, 91% of hospitals routinely monitored antimicrobial shortages, and all had systems to report substandard and falsified medicines. Training initiatives also improved, with about 82% of facilities providing AMS and IPC training. Implementation of AMS and IPC shows positive outcomes in health facilities, as has been shown in previous studies [60,61]. However, fully implemented prescribing optimization programs and ongoing in-service training remained limited. Hence, this affected the comprehensive execution of AMS actions in the hospitals. The findings are similar to the findings for hospitals in the following countries [62,63]. These findings show the importance of continuous educational interventions and system improvements in hospitals.

Overall, while the results indicate meaningful progress in AMS program implementation, particularly in governance structures, critical gaps remain in functional domains such as clinical audit-feedback, prescribing optimization, and financial sustainability. The findings underscore the need for continued capacity-building, resource mobilization, and institutional accountability with monitoring frameworks that can translate AMS interventions into measurable clinical outcomes. This is because effective AMS implementation in hospitals would reduce the prevalence of AMU and AMR, promote appropriate use of antimicrobials, improve awareness, knowledge, perceptions, and practices regarding the use of antimicrobials [18,20,23,24,49,52,59,64,65,66,67]. Therefore, strengthening AMS in Zambian hospitals is critical to combat AMR, similar to findings reported in other LMICs [68].

We are aware that this study has some limitations that should be considered when interpreting the findings. First, although the WHO-adapted assessment tool was reviewed and validated for use in Zambia, it remains a self-reported measure, which may introduce reporting bias, especially where respondents may overstate progress. Second, the study was conducted in 11 sentinel public hospitals with relatively better infrastructure and resources compared to many other facilities in Zambia, which may limit the generalizability of the findings to lower-level or rural health facilities. Third, the implementation period was 12 months, which may not have been sufficient to capture the full impact or sustainability of AMS interventions, particularly in areas requiring cultural or systemic change. Fourth, while improvements in AMS core element scores were documented, the study did not directly measure clinical outcomes such as patient morbidity, mortality, or antimicrobial resistance trends, limiting the ability to link program improvements to patient-level benefits. Fifth, variability in resources, leadership commitment, and technical capacity across facilities may have influenced both the implementation process and the observed results. Finally, the absence of a formal control group and the reliance on a pre–post design mean that other concurrent initiatives or contextual factors cannot be ruled out as contributors to the observed changes.

However, this study provides one of the first comprehensive assessments of AMS implementation in Zambia’s public hospitals using a standardized, WHO-adapted tool. By demonstrating that targeted, multidisciplinary interventions can strengthen AMS core elements within a short timeframe, the findings offer actionable evidence for policymakers, hospital leaders, and development partners. The study’s insights can guide national scale-up strategies, inform resource allocation, and serve as a model for other low- and middle-income countries seeking to institutionalize AMS and address antimicrobial resistance within constrained health systems.

### Policy Recommendations and Implications

The study highlights the need for comprehensive policy action to strengthen AMS in Zambia’s public hospitals, as shown in Table 3. Key priorities include reinforcing governance through dedicated leadership and functional committees, securing sustainable funding, and building workforce capacity via continuous training. Strengthening core AMS actions, ensuring regular monitoring and feedback, institutionalizing DTC operations, and improving supply chain and laboratory support are critical to sustaining gains. Integrating AMS into broader health policies and advocacy efforts will further enhance national capacity to combat AMR.

## 4. Materials and Methods

### 4.1. Study Sites and Design

We conducted cross-sectional studies at the baseline (pre-intervention) and endline (post-intervention) periods across eleven (11) hospitals located in the ten provinces of Zambia. This was an implementation-strengthening study across hospitals at different baseline capacities. The period between the assessments was twelve (12) months, spanning from September 2023 to October 2024. The hospitals included both provincial referral and designated AMR surveillance hospitals, namely Livingstone Teaching Hospital (LTH), Ndola Teaching Hospital (NTH), University Teaching Hospital (UTH), Lewanika General Hospital (LGH), Chipata Central Hospital (CCH), Kabwe Central Hospital (KCH), Mansa General Hospital (MGH), Chinsali General Hospital (CnGH), Kasama General Hospital (KGH), Choma General Hospital (CGH), and Solwezi General Hospital (SGH) (Figure 6). These were tertiary and secondary level hospitals that had established AMS committees after baseline AMS surveys. Compared to other hospitals in the country, these institutions possess relatively better infrastructure to support AMS programs, including the capacity to perform a range of microbiology laboratory tests from basic to advanced, as previously described [34].

### 4.2. Target Population and Sampling Technique

We conducted interviews with at least four members of the AMS team, DTC, or ICC in each healthcare facility using the modified Periodic National and Healthcare Facility Assessment Tool in the WHO policy guidance on integrated AMS activities in human health for both the baseline and endline assessments. The individuals selected for these interviews were purposefully chosen due to their direct involvement in implementing AMR and AMS activities within their respective healthcare facilities. The interview team comprised multidisciplinary staff, including environmental technologists, pharmacists, nurses, clinicians, and laboratory scientists and technicians. Given the implementation nature of the study, no formal sample size formula using alpha and beta was applied. Therefore, all eligible sentinel hospitals were included for maximum coverage. As this was a census of all 11 selected hospitals and all participated in both assessments, there was no dropout. The purposive sampling strategy ensured representativeness of AMR sentinel facilities, including hospitals from all 10 provinces and both provincial and tertiary levels.

### 4.3. Data Collection

The modified Periodic National and Healthcare Facility Assessment Tool in the WHO policy guidance on integrated AMS activities in human health [69] was used for data collection for both the baseline and end-of-period one assessments. Before its initial use, the WHO-adapted tool was reviewed, adapted, and validated by a multidisciplinary stakeholder group for use in Zambia’s healthcare settings. The tool was administered in English, the official language used in all hospitals. The tool is designed for periodic assessments at baseline, period one, period two, and subsequent phases, allowing for continuous monitoring of the AMS program progress. Hence, the tool facilitates the assessment of the AMS’s six core elements at different intervals. It acts as a monitoring tool for the AMS program and activity implementation. It evaluates six core elements: leadership commitment; accountability and responsibility; AMS actions; education and training; monitoring and surveillance; and reporting and feedback [69]. Additionally, indicators related to governance structures, including Drug and Therapeutics Committees (DTCs), Infection Control Committees (ICCs), and AMS teams, were also assessed [38,69].

The facility’s AMS program performance was categorized based on percentage scores. Programs achieving between 80% and 100% were considered fully implemented and functional, though requiring sustained support for continuity [34]. Scores between 50% and 79.9% indicated partial functionality, signaling the need for targeted improvements. Scores below 50% identified facility AMS programs that were either poorly functioning or non-operational and in need of prioritized intervention [34,69]. While the individual indicators were evaluated using a scoring system that ranged from 0 to 4. A score of 0 indicated no implementation; 1 indicated the action was not in place but was prioritized; 2 denoted that the action was planned but not initiated; 3 reflected partial implementation; and 4 represented full implementation [34,69].

The data collection team comprised members of the 33 (three per facility) AMS National Technical Working Group (TWG), all of whom received training on administering the assessment tool. To enhance consistency and minimize bias during interviews, role-playing sessions were conducted. Data was gathered using tablets and computers, with access limited to authorized users.

### 4.4. Implementation

Recognizing the importance of AMS, the Zambia National Public Health Institute (ZNPHI), through the Antimicrobial Resistance Coordinating Committee (AMRCC), initiated a nationwide AMS program assessment at selected AMR sentinel sites. The baseline (pre-implementation) assessment focused on evaluating healthcare facility compliance with the WHO AMS core elements, identifying policy gaps, and understanding structural and operational barriers to effective stewardship. Following the identification of these gaps, targeted interventions were rolled out, including staff training, IPC enhancement, and support for Drug and Therapeutics Committees (DTCs). After a 12-month implementation period, a follow-up (post-implementation: the 12-month follow-up assessment conducted after implementing targeted AMS intervention) assessment was conducted to evaluate progress and impact of AMS interventions.

The baseline assessment revealed the gaps and guided the focus areas for intervention implementation. Several interventions in these healthcare facilities were implemented. These included the development and implementation of guidance documents such as the AMS guidelines, antibiotic prescription charts, local antibiograms, and AMS training manuals. Review of the treatment guidelines and essential medicines list, AMS ward rounds, and peer-to-peer technical supportive visits. The other activities involved orientation of the hospital leadership as a way of building ownership of the program, training of facility staff in AMS and AMR, and monthly facility AMS committee meetings. Further, conducted regular point prevalence surveys on antibiotic use, prescription audits, and strengthening of the supply chain for laboratory commodities and antimicrobials. Moreover, onsite mentorships, virtual AMS Echo sessions, and AMS and data review meetings were held to standardize the implementation and improve the quality of the data generated (Figure 7).

### 4.5. Data Analysis

Data collected using the modified Periodic National and Healthcare Facility Assessment Tool in the WHO policy guidance on integrated AMS activities in human health documents were cleaned and aggregated. There were no missing responses in the final dataset, as all tools were checked for completeness during collection. As the tool is self-scoring, it automatically generates summary scores expressed as percentages for each core element. These results were interpreted as previously described.

Additionally, statistical analyses were performed using Excel and R Studio Version 2024.12.1+563. Further, the core elements in which healthcare facilities scored the lowest in the baseline assessment, especially DTC functionality, AMS actions, and reporting and feedback, were analyzed to observe the changes in the period one assessment. Figure 7 shows the workflow in this study from baseline to period one assessments.

## 5. Conclusions

This study demonstrates that a multidisciplinary, targeted approach to implementing Antimicrobial Stewardship (AMS) programs can lead to measurable improvements in AMS core element performance in Zambia’s public hospitals, despite variations in baseline capacity and resource availability. The observed progress underscores the importance of strengthening governance structures, securing sustainable financing, enhancing workforce capacity, improving supply chain and laboratory support, and embedding AMS into national health policies. While notable gains were achieved within a relatively short period, sustained impact will require long-term commitment, continuous monitoring, and adaptive strategies tailored to facility contexts. By institutionalizing AMS as a core component of health service delivery, Zambia can not only mitigate the growing threat of antimicrobial resistance but also contribute to global efforts in safeguarding the effectiveness of life-saving medicines.

## Figures and Tables

**Figure 1 antibiotics-14-01125-f001:**
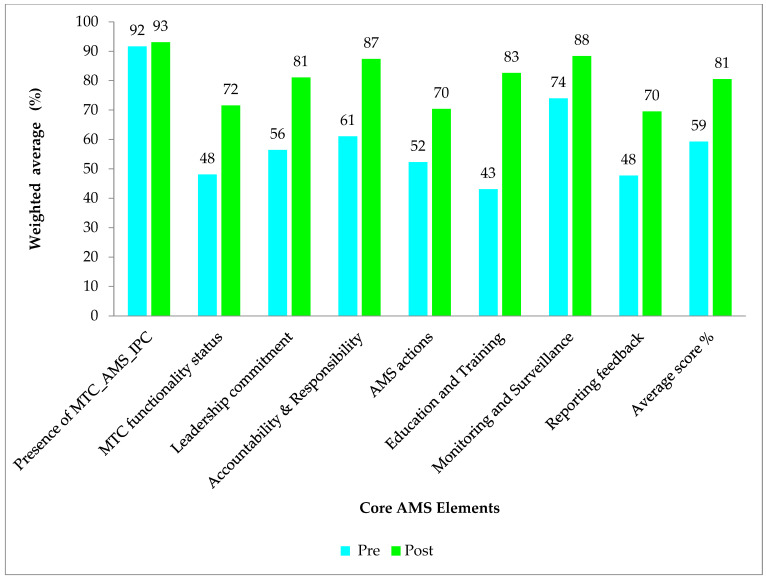
The figure shows the average scores in the AMS core elements (%) for baseline and period one assessment.

**Figure 2 antibiotics-14-01125-f002:**
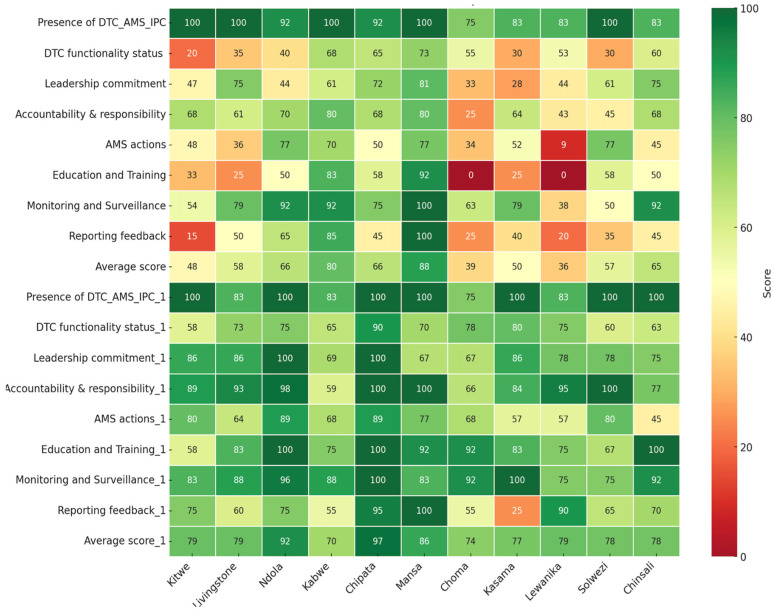
Heat map indicating the scores for the baseline and period one assessment of AMS implementation across eleven hospitals in Zambia.

**Figure 3 antibiotics-14-01125-f003:**
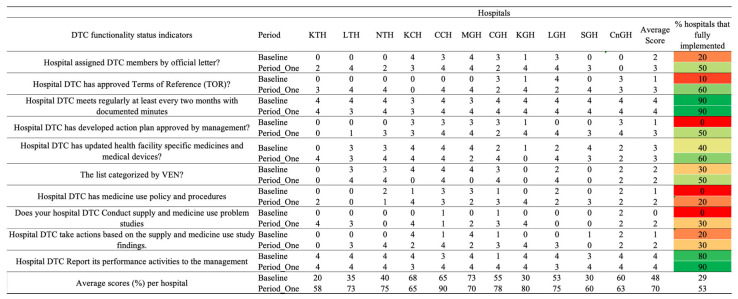
Baseline and period one assessment scores for Drug and Therapeutics Committee (DTC) functionality indicators across 11 hospitals in Zambia. Scores are presented using a standardized 0–4 scale, with corresponding percentage performance. A score of 0 = no implementation; 1 = action was not in place but was prioritized; 2 = planned but not initiated; 3 = partial implementation; and 4 = full implementation. Color coding reflects implementation levels: red (poor), orange/yellow (partial), and green (good).

**Figure 4 antibiotics-14-01125-f004:**
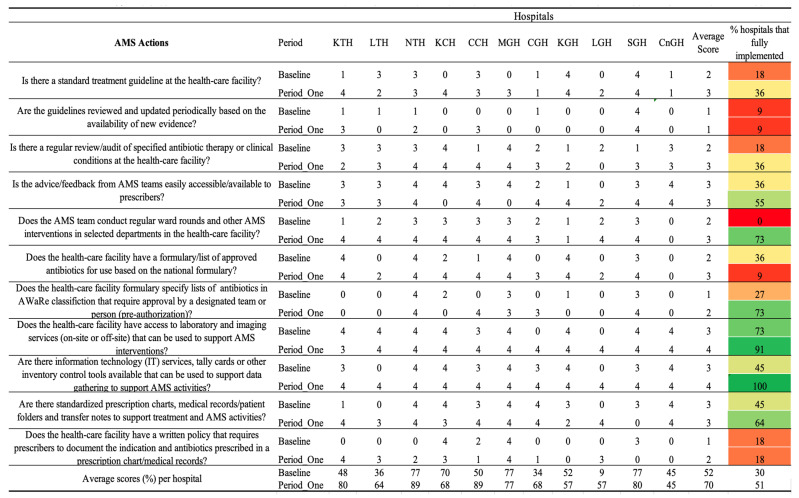
Baseline and period one assessment scores for AMS Actions across 11 hospitals in Zambia. Scores are presented using a standardized 0–4 scale, with corresponding percentage performance. A score of 0 = no implementation; 1 = action was not in place but was prioritized; 2 = planned but not initiated; 3 = partial implementation; and 4 = full implementation. Color coding reflects implementation levels: red (poor), orange/yellow (partial), and green (good).

**Figure 5 antibiotics-14-01125-f005:**
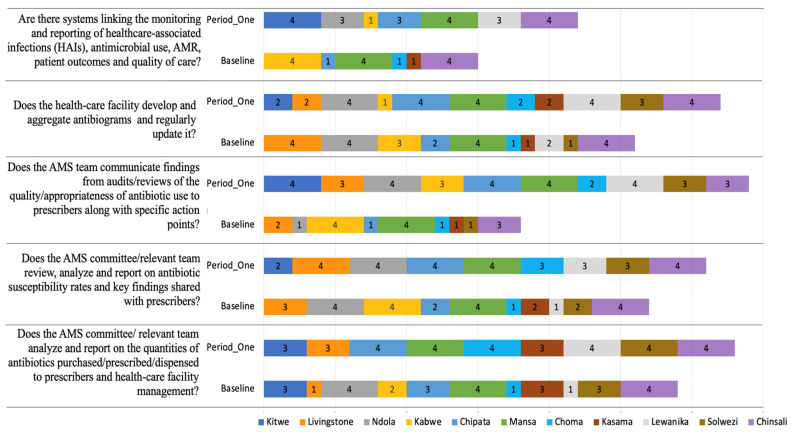
Baseline and Period One assessment scores for Reporting and Feedback indicators across 11 hospitals in Zambia. The figure illustrates the implementation scored from 0 to 4. A score of 0 = no implementation; 1 = action was not in place but was prioritized; 2 = planned but not initiated; 3 = partial implementation; and 4 = full implementation. Facilities are color-coded, comparing baseline and follow-up assessments to highlight progress and persistent gaps.

**Figure 6 antibiotics-14-01125-f006:**
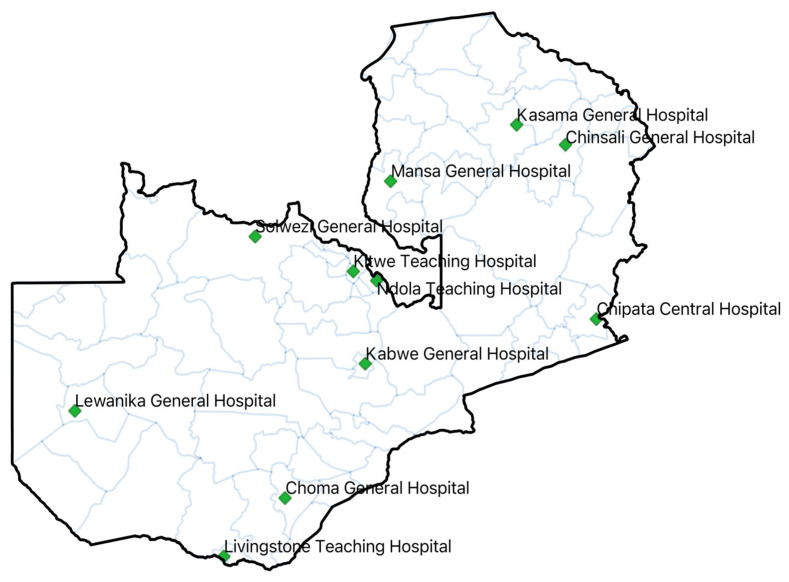
Map of Zambia with hospitals in which baseline and period one assessments on AMS were conducted.

**Figure 7 antibiotics-14-01125-f007:**
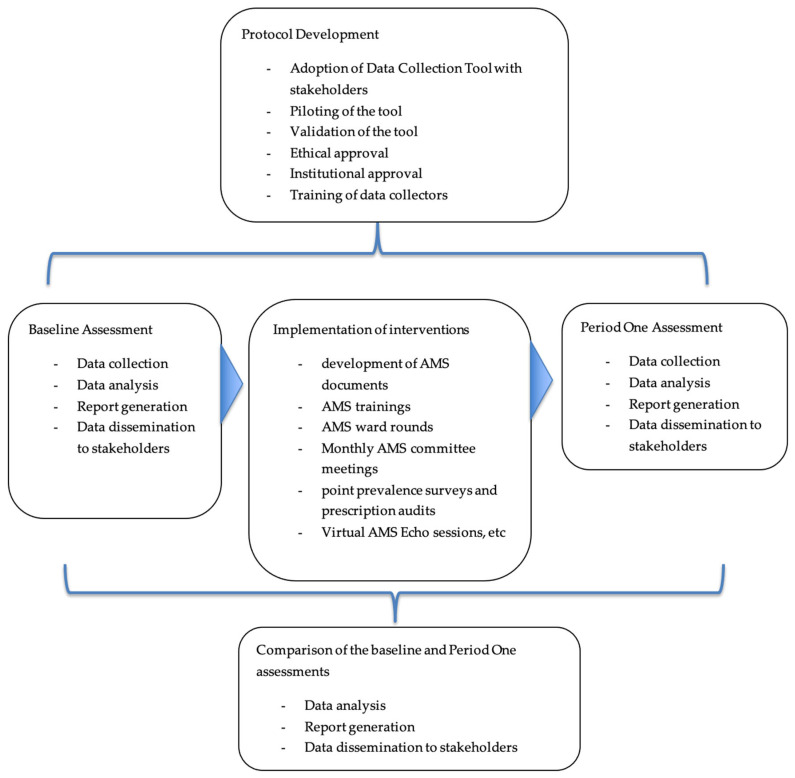
Workflow for Baseline and Period One Assessments for Antimicrobial Stewardship Program.

**Table 1 antibiotics-14-01125-t001:** Demographics of Public Hospitals Assessed at Baseline and Follow-up Periods.

Level of Care	Public Hospital Name	Province	No. of Bed Spaces	Hospital Annual Admissions
Tertiary	Ndola Teaching	Copperbelt	741	19,656
	Kitwe Teaching	Copperbelt	500	15,101
	Arthur Davidson Children’s Teaching	Copperbelt	250	10,800
	Livingstone Teaching	Southern	325	10,871
	Chipata Central	Eastern	600	11,620
	Kabwe Central	Central	474	14,940
Secondary	Mansa General	Luapula	420	4908
	Chilonga Mission General	Muchinga	230	3500

**Table 2 antibiotics-14-01125-t002:** Demographic Characteristics of Study Participants.

Category	Frequency	Percent
Gender		
Female	20	63
Male	12	38
Profession		
Pharmacists	8	25
Environmental Technologist	3	9
Clinicians	8	25
Nurses	6	19
Laboratory Personnel	7	22

**Table 3 antibiotics-14-01125-t003:** Policy Recommendations to Strengthen Antimicrobial Stewardship (AMS) Implementation in Zambia’s Public Hospitals.

Policy Area	Recommendation	Key Actions
Governance and Leadership	Strengthen hospital-level AMS governance structures	• Appoint and train dedicated AMS focal persons in all hospitals • Integrate AMS into annual facility action plans with measurable indicators • Establish clear terms of reference for AMS committees
Financing and Sustainability	Secure sustainable funding for AMS programs	• Create dedicated budget lines for AMS activities at the facility and national levels • Reduce reliance on donor funding by advocating for domestic resource mobilization • Incorporate AMS into national health financing strategies
Human Resources and Capacity Building	Enhance the AMS team capacity through continuous training	• institutionalize regular in-service training for all healthcare workers on AMS/IPC • Include AMS competencies in pre-service curricula for pharmacists, nurses, and clinicians • Develop mentorship and peer-learning networks between high- and low-performing hospitals
AMS Actions and Clinical Practice	Improve implementation of core AMS actions	• Ensure regular AMS ward rounds in all facilities • Develop and periodically update local treatment guidelines and antibiograms • Implement mandatory documentation of clinical indication and antibiotic choice
Monitoring, Feedback and Reporting	Strengthen data-driven AMS decision-making	• Establish regular audit-feedback cycles on antibiotic use and AMS indicators • Link AMS data with HAI and AMR surveillance • Ensure timely dissemination of reports to prescribers and hospital leadership
Drug and Therapeutics Committees (DTCs)	institutionalize functional DTC operations	• Hold monthly DTC meetings with documented action points • Develop medicine use policies and procedures • Conduct periodic drug use problem studies and address findings
Supply Chain and Laboratory Support	Ensure uninterrupted access to essential AMS resources	• Strengthen procurement and stock monitoring systems for antimicrobials, lab commodities, and PPE • Expand laboratory capacity for culture and susceptibility testing • Integrate AMS needs into national procurement frameworks
Policy Integration and Advocacy	Align AMS with broader health policies and programs	• Embed AMS in national quality improvement, IPC, and UHC frameworks • Advocate for AMS in national health strategies and donor-funded health programs • Promote public awareness campaigns on AMR and prudent antibiotic use

## Data Availability

The original contributions presented in the study are included in the article/Appendix A; further inquiries can be directed to the corresponding author.

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
