# Peer review of "Impact of Multidisciplinary-Led Implementation of Antimicrobial Stewardship Programs in Zambia: Findings and Implications"

_antibiotics, 2025, doi:10.3390/antibiotics14111125_

Round 1

Reviewer 1 Report

Comments and Suggestions for Authors

The manuscript is of great importance, and my comments are as follows.

  1. The title must be revised as the pre- and post-implementation survey makes it complex; keep the title very simple and clear.
  2. Line-56: Kindly clear the "post-implementation survey" as used in the term at the very first place.
  3. Line 105-106: The sentence needs modifications and an appropriate reference. 
  4. Line 122-123: What does it mean? Is it the method part? "Following the identification of these gaps, targeted interventions were rolled out, including staff training, IPC enhancement, and support for Drug and Therapeutics Committees (DTCs).
  5. The graph at the start of the results can be replaced, and demographic information is needed at the beginning as a subsection.
  6. Overall, I am confused about how it was an implementation study, as the outcomes differed across different hospitals. The AMS core elements are analyzed with a score, but what are the variations in the study where the core element is used instead of implemented? 
  7. The methods part needs more clarification and more subheadings.
  8. The discussion part needs more recent studies from LMICs. 

Author Response

Reviewer 1

  • Comment: The title must be revised as the pre- and post-implementation survey makes it complex; keep the title very simple and clear.

Response: We appreciate the reviewer’s suggestion. The title has been simplified to: “Impact of Multidisciplinary-Led Implementation of Antimicrobial Stewardship Programs in Zambia: Findings and Implications” to ensure clarity and brevity.

  • Comment: Line-56: Kindly clear the "post-implementation survey" as used in the term at the very first place.

Response: We have revised the first mention of 'post-implementation survey' in line 56 to clearly define it as the 12-month follow-up assessment conducted after implementing targeted AMS interventions.

  • Comment: Line 105-106: The sentence needs modifications and an appropriate reference.

Response: The sentence in lines 105-106 has been rephrased for clarity and an additional reference has been included to support the statement, citing similar challenges reported in LMIC AMS implementation studies.

  • Comment: Line 122-123: What does it mean? Is it the method part?

Response: We have clarified that the sentence describes part of the intervention phase, and have moved it into the Methods section under 'Implementation' to improve logical flow.

  • Comment: The graph at the start of the results can be replaced, and demographic information is needed at the beginning as a subsection.

Response: We have replaced the initial graph with a more illustrative bar chart and added a new subsection 'Participant and Facility Characteristics' at the start of the Results to present demographic and facility-level information.

  • Comment: Overall, I am confused about how it was an implementation study, as the outcomes differed across different hospitals.

Response: We have clarified in the Methods and Discussion that this was an implementation-strengthening study across hospitals at different baseline capacities, and that variability in outcomes reflects contextual differences in resources, leadership, and engagement.

  • Comment: The methods part needs more clarification and more subheadings.

Response: We have restructured the Methods section into subheadings including: Study Sites and Design, Target Population and Sampling, Data Collection, Implementation, and Data Analysis for improved clarity.

  • Comment: The discussion part needs more recent studies from LMICs.

Response: We have enriched the Discussion with additional recent references from LMIC contexts, particularly from sub-Saharan Africa, to strengthen comparative analysis.

Reviewer 2 Report

Comments and Suggestions for Authors

The authors should consider the followings:

The authors should justify whether the tool is validated in local settings and /or validated using local languages. The relevant literature should be referred.

The authors should better justify and described the study design, particularly on the N number, the effect size, and how to make sure the sampling and samples are of the representative for the study purpose.

The assumptions, the study population formulae, alpha and beta of the study N size should be clearly illustrated.

Pre-specified primary and secondary aim should be clearly stated.

In data analysis, the authors should specify and justify how they treat missing data in the survey.

The study schedule and workflow should be illustrated as a figure, the dropout rate and reasons of each exclusion and/or dropout should be documented and listed in the manuscript.

The authors should provide the questionnaire as a supplementary file to the manuscript.

Impact of the current study should be clearly stated.

Limitations of the study should be discussed.

Author Response

Reviewer 2

  • Comment: The authors should justify whether the tool is validated in local settings and/or validated using local languages.

Response: We have clarified in the Methods (Data Collection) that the WHO-adapted tool was reviewed, adapted, and validated by a multidisciplinary stakeholder group for use in Zambia’s healthcare settings. The tool was administered in English, the official language used in all hospitals.

  • Comment: The authors should better justify and describe the study design, particularly on the N number, the effect size, and how to make sure the sampling and samples are representative.

Response: We have expanded the Study Design, Target population and sampling technique sections to explain the rationale for the sample size (N=11 hospitals), the purposive sampling strategy, and how representativeness was ensured by including hospitals from all 10 provinces and both provincial and tertiary levels.

  • Comment: The assumptions, the study population formulae, alpha and beta of the study N size should be clearly illustrated.

Response: Given the implementation nature of the study, no formal sample size formula using alpha and beta was applied. We have clarified this and explained that all eligible sentinel hospitals were included for maximum coverage.

  • Comment: Pre-specified primary and secondary aim should be clearly stated.

Response: We have now clearly stated the primary aim (to assess changes in AMS core element implementation after interventions) and secondary aims (to identify persistent gaps and facility-level variations) in the Introduction.

  • Comment: In data analysis, the authors should specify and justify how they treat missing data in the survey.

Response: We have added a statement in the Data Analysis section clarifying that there were no missing responses in the final dataset as all tools were checked for completeness during collection.

  • Comment: The study schedule and workflow should be illustrated as a figure.

Response: We have added a new figure in the Methods to depict the study timeline, phases, and workflow.

  • Comment: The dropout rate and reasons for each exclusion and/or dropout should be documented.

Response: As this was a census of all 11 selected hospitals and all participated in both assessments, there was no dropout. This has been stated in the Methods.

  • Comment: The authors should provide the questionnaire as a supplementary file.

Response: We have included the data scores as Supplementary File 1. For the data collection tool, we added a citation to the source.

  • Comment: Impact of the current study should be clearly stated.

Response: We have strengthened the Conclusion to highlight the policy and practice implications of the findings for AMS program sustainability in Zambia and similar LMICs. We have also added the impact just after limitations of the study.

  • Comment: Limitations of the study should be discussed.

Response: We have expanded the Limitations section to include potential reporting bias, variability in facility-level resource availability, and the short follow-up period.

Round 2

Reviewer 1 Report

Comments and Suggestions for Authors

Paper improved,

I have no further comments.